# Intelligent Control with Artificial Neural Networks for Automated Insulin Delivery Systems

**DOI:** 10.3390/bioengineering9110664

**Published:** 2022-11-08

**Authors:** João Lucas Correia Barbosa de Farias, Wallace Moreira Bessa

**Affiliations:** 1Department of Mechanical Engineering, Federal University of Rio Grande do Norte, Natal 59078-970, RN, Brazil; 2Department of Mechanical and Materials Engineering, University of Turku, 20500 Turku, Finland

**Keywords:** artificial pancreas, automated insulin delivery, blood glucose regulation, intelligent control, radial basis functions, neural networks

## Abstract

Type 1 diabetes *mellitus* is a disease that affects millions of people around the world. Recent progress in embedded devices has allowed the development of artificial pancreas that can pump insulin subcutaneously to automatically regulate blood glucose levels in diabetic patients. In this work, a Lyapunov-based intelligent controller using artificial neural networks is proposed for application in automated insulin delivery systems. The adoption of an adaptive radial basis function network within the control scheme allows regulation of blood glucose levels without the need for a dynamic model of the system. The proposed model-free approach does not require the patient to inform when they are going to have a meal and is able to deal with inter- and intrapatient variability. To ensure safe operating conditions, the stability of the control law is rigorously addressed through a Lyapunov-like analysis. In silico analysis using virtual patients are provided to demonstrate the effectiveness of the proposed control scheme, showing its ability to maintain normoglycemia in patients with type 1 diabetes *mellitus*. Three different scenarios were considered: one long- and two short-term simulation studies. In the short-term analyses, 20 virtual patients were simulated for a period of 7 days, with and without prior basal therapy, while in the long-term simulation, 1 virtual patient was assessed over 63 days. The results show that the proposed approach was able to guarantee a time in the range above 95% for the target glycemia in all scenarios studied, which is in fact well above the desirable 70%. Even in the long-term analysis, the intelligent control scheme was able to keep blood glucose metrics within clinical care standards: mean blood glucose of 119.59 mg/dL with standard deviation of 32.02 mg/dL and coefficient of variation of 26.78%, all below the respective reference values.

## 1. Introduction

Diabetes *mellitus* is a metabolic disease characterized by hyperglycemia resulting from defects in insulin action, secretion, or both. Chronic hyperglycemia caused by diabetes is associated with long-term damage and failure of several organs, such as kidneys, eyes, heart and nerves [1]. The two more common types of diabetes are Type 1 diabetes *mellitus* (T1DM) and Type 2 diabetes *mellitus* (T2DM). T1DM consists of an autoimmune destruction of pancreatic β-cells, which are responsible for producing insulin, the hormone that lowers blood glucose concentration (BGC). In this case, the subcutaneous administration of insulin is indeed crucial for survival.

For patients with T1DM, modern therapy consists of continuous blood glucose monitoring (CGM) and automatic insulin delivery. CGM devices, which measure interstitial glucose concentration, have great benefits for diabetes management [2]. By combining a CGM device with an insulin pump, it became possible to deploy automated insulin delivery (AID) systems, also known as an artificial pancreas. However, in addition to the insulin pump and the CGM sensor, an appropriate algorithm to automatically define the amount of insulin to be delivered to the patient is also a fundamental part of developing an effective artificial pancreas.

Several control strategies have been proposed for automatic blood glucose regulation. Basically, they can be categorized as linear or nonlinear and model-based or model-free techniques. Proportional–integral–derivative (PID), fuzzy inference and model predictive control (MPC) are the most common approaches for AID systems, being the only ones with clinical validation so far [3,4,5]. MPC has been extensively used [6,7,8,9] and, when compared PID controllers, show better performance [10,11]. Nevertheless, Haidar [12] shows that fuzzy logic can still improve MPC performance and reduce the risk of hypoglycemia, which indicates the relevance of computational intelligence for the design of artificial pancreas.

Fuzzy logic, like many other artificial intelligence (AI) algorithms, allows the development of model-free controllers, that is, without the need for a patient model. This can be quite useful, as we are dealing with a biological system which is highly complex, nonlinear, uncertain and subject to external disturbances and time delays. In fact, obtaining mathematical models that can adequately capture human physiological behavior for use in control schemes has been a major challenge [13]. Vettoretti et al. [14] stress that, due to the huge amount of data collected from T1DM patients using CGM, AI techniques can give support to bolus insulin calculation [15] and BGC prediction [16,17,18,19], that is, predict future glycemic profiles based on present and past values of glycemia. Furthermore, reinforcement learning (RL) has also been applied in the context of AID systems with an increasing interest in the past few years [20,21,22]. In [23], a bioinspired RL algorithm was used to automate insulin infusion, by selecting reward functions that imply temporal BGC homeostasis.

For example, the main sources of nonlinearity in blood glucose dynamics come from the insulin action on parameters of glucose production, glucose distribution/transport and glucose disposal [24]. On this basis, due to the need to linearize the plant around an operating point, the adoption of linear controllers may lead to several issues in blood glucose regulation. This has led in recent years to the development of nonlinear control strategies for AID systems.

Nonlinear MPC was tested in a clinical study with 10 T1DM patients [24], showing satisfactory results in maintaining normoglycemia and reducing hypoglycemic events. In addition, feedback linearization (FBL) [25] and sliding mode control (SMC) [26,27] have been also used in the design controllers for artificial pancreas. Abu-Rmileh and Garcia-Gabin [28] combine SMC and a nonlinear Wiener model for in silico analysis, resulting in faster response for food intake when compared to MPC controller. Moreover, Ahmad et al. [29] and Ebrahimi et al. [30] also implemented SMC controllers for artificial pancreas and obtained sound results. Nath et al. [31], in turn, developed a nonlinear adaptive controller that is able to provide desired glycemic control under uncertain meal disturbance and intrapatient variability.

It is also worth noting that the use of subcutaneous route to deliver insulin to the patient adds significant delay to the system’s dynamics. Furthermore, the unidirectional effect of insulin on glucose dynamics may hamper the safe and efficient regulation of the BGC because the controller cannot counteract the effects of insulin overdosing [13]. Thus, to improve control performance, many AID systems are also designed with state, parameter or uncertainty estimators to deal with both external disturbances and unmodeled dynamics. State estimation has been performed in [32,33,34] to aid the controller and improve its overall performance. Due to the great inter and intrapatient variability, some control schemes consider parameter estimation to improve plant modeling and, consequently, control performance [7,33,35].

An essential feature in modern AID systems is the ability to maintain postprandial normoglycemia without the need to report meals to the device. Typically, the patient should inform the insulin delivery system about a meal and the amount of carbohydrate in it. Meal announcement and carbohydrate calculation, which is usually not accurate, are major concerns for patients and often restrict their eating. In order to attain this automatically, a separate module may be used to detect food intake [36,37,38,39,40].

However, most control schemes proposed so far for automated insulin delivery face severe limitations: time-delays, for example, may compromise the stability of PID controllers [41]; model predictive control is often subject to excessive computational burden due to its associated optimization issues [42]; fuzzy control is typically heuristic and generally lacks a systematic framework and rigorous tools for stability analysis [43]; sliding mode controllers may suffer from chattering issues and need a reasonable plant model as well as full state feedback [44].

In this work, we introduce a novel intelligent controller for the automatic regulation of blood glucose levels. For this purpose, an artificial neural network (ANN) is incorporated in a Lyapunov-based nonlinear control law to ensure the stability of the closed-loop system. Online learning, instead of offline training, is proposed to update the ANN weights, which allows continuous learning and manages to cover all the nuances and variability of human physiology. Following [45,46], the projection algorithm [47] is also employed to ensure that the ANN weights remain within predefined bounds, in order to avoid hypoglycemia due to insulin overdosing. By combining the artificial neural network with a non-linear control scheme, the computational demands on the ANN are drastically reduced. This in fact scales down the computational complexity of the algorithm and allows the proposed controller to be deployed on the low-power embedded hardware of automated insulin delivery systems. The efficacy of the proposed control scheme is evaluated through numerical simulations using the Identifiable Virtual Patient (IVP) model presented in [48]. It should be emphasized that the IVP model is used for simulation purposes only, but is not considered in the design of the control law.

The main advantages of the control approach introduced in this work are summarized below:It requires only one state, namely the blood glucose concentration, which is measured once every five minutes;No dynamic model of the patient is needed for the design of the control law;It is able to maintain normoglycemia (during postprandial and fasting periods) without the need for meal announcement or carbohydrate calculation;It is able to deal with both inter- and intrapatient variability;It has a good trade-off between simplicity and efficacy, which allows its implementation on low-power embedded hardware.

The rest of this paper is organized as follows: Section 2 presents the virtual patient model used for simulation purposes (only) during in silico analysis; Section 3 introduces the proposed intelligent controller with the corresponding Lyapunov-like stability analysis to demonstrate the boundedness and convergence properties of all closed-loop signals; Section 4 presents the employed materials and methods; the numerical results obtained in both short- and long-term simulations are presented and discussed in Section 5; and Section 6 summarizes the concluding remarks.

## 2. Virtual Patient Model

In order to evaluate the control scheme by means of computer simulations, a virtual patient is needed to emulate the physiology of a real one. The virtual patient is a mathematical model that describes blood glucose dynamics, that is, how BGC changes over time under the influence of food intake and variations in both glycemia and plasma insulin concentration. The Identifiable Virtual Patient model proposed by Kanderian et al. [48] is then considered here for in silico analysis. The IVP is based on two previous models [24,49] and has already been used in [50] for the controller design and in [51] for the design of state, parameter and disturbance estimators in artificial pancreas.

The IVP model can be described as a four-dimensional vector field:(1)$$\begin{array}{cc} \; & {\dot{I}}_{\mathrm{SC}}\left(t\right)=-\frac{I_{\mathrm{SC}}\left(t\right)}{\tau_{1}}+\frac{u(t)}{\tau_{1}C_{I}}\\ \; & \dot{I}\left(t\right)=\frac{I_{\mathrm{SC}}\left(t\right)}{\tau_{2}}-\frac{I(t)}{\tau_{2}}\\ \; & {\dot{I}}_{\mathrm{EFF}}\left(t\right)=-p_{2}I_{\mathrm{EFF}}\left(t\right)+p_{2}S_{I}I\left(t\right)\\ \; & \dot{G}\left(t\right)=-\left[\mathrm{GEZI}+I_{\mathrm{EFF}}\left(t\right)\right]G\left(t\right)+\mathrm{EGP}+R_{A}\left(t\right) \end{array}$$
where $I_{\mathrm{SC}}\left(t\right)$ and I(t) are subcutaneous and plasma insulin concentrations [μU/mL], respectively; $I_{\mathrm{EFF}}\left(t\right)$ is the insulin effect on blood glucose [min${}^{-1}$]; G(t) represents the blood glucose concentration [mg/dL]; u(t) stands for the subcutaneous insulin infusion [μU/min], which is equivalent to the control input; and $R_{A}\left(t\right)$ represents the rate of glucose appearance [mg/dL/min], i.e., the increase in blood glucose due to food intake, which can be described by
(2)$$R_{A}\left(t\right)=\frac{C_{H}\;t}{V_{G}\;\tau_{m}^{2}}\;\exp\left(-t/\tau_{m}\right)$$
with $C_{H}$ being the amount of carbohydrate ingested [mg], $V_{G}$ the glucose distribution volume [dL], and $\tau_{m}$ the peak time of meal glucose appearance [min].

The other IVP parameters are the pharmacokinetic time constants $\tau_{1}$ and $\tau_{2}$ [min], the insulin clearance $C_{I}$ [mL/min], the time constant for insulin action $p_{2}$ [min${}^{-1}$], the insulin sensitivity $S_{I}$ [mL/μU/min], the glucose effectiveness at zero insulin GEZI [min${}^{-1}$], and the endogenous glucose production EGP [mg/dL/min].

## 3. Intelligent Control

In view of its ability to adapt itself, learn from experience and adequately predict plant dynamics [52], intelligent control can be considered a very suitable approach for healthcare applications. Intelligent controllers are capable of handling highly uncertain systems and have been successfully employed in robotics [45,52] and many other mechatronic systems [53,54].

In order to design the model-free intelligent controller, let us start by assuming that the plant dynamics can be described by an uncertain first-order nonlinear system:(3)$$\dot{x}\left(t\right)=f\left(x,t\right)+b\left(x,t\right)\;u\left(t\right)$$
where x(t)=G(t) represents the blood glucose concentration, *f* and *b* are both nonlinear functions, and u(t) is the control input.

It is important to highlight that, by choosing the first-order system (Equation 3) to represent all the physiological dynamics of the patient with T1DM, it is necessary to ensure that the controller is capable of dealing with the disturbances/uncertainties arising from food intake and insulin-glucose metabolism.

Thus, considering that (Equation 3) is subject to external disturbances as well as modeling uncertainties, i.e., $f=\hat{f}+\Delta f$ and $b=\hat{b}+\Delta b$, it can be rewritten as follows:(4)$$\dot{x}\left(t\right)=\hat{f}\left(x,t\right)+\hat{b}\left(x,t\right)\;u\left(t\right)+d$$
where *d* comprises the disturbances due to food intake and modeling uncertainties, Δf and Δb.

Recalling that radial basis function (RBF) neural networks with a single hidden layer can ensure universal approximation [55,56], we propose the adoption of a RBF network with *m* neurons in the hidden layer (see Figure 1) to cope with the uncertainties/disturbances that can arise:(5)$$\hat{d}=w^{⊤}\varphi \left(x\right)$$
where w is the weight vector and φ(x) contains the activation functions of each neuron in the hidden layer, i.e.,
(6)$$\varphi \left(x\right)={\left[\varphi_{1}\left(x\right)\;\varphi_{2}\left(x\right)\;\cdots \;\varphi_{m}\left(x\right)\right]}^{⊤}$$

RBF networks have been successfully employed for uncertainty/disturbance compensation in the intelligent control of many uncertain complex systems [46,52].

Now, we evoke the feedback linearization method [57] to design an appropriate nonlinear control law for the uncertain first-order nonlinear system (Equation 4):(7)$$u={\hat{b}}^{-1}\left(-\hat{f}+{\dot{x}}_{d}-\lambda \tilde{x}-\hat{d}\right)$$
where $x_{d}$ is the desired blood glucose concentration (target glycemia), $\tilde{x}=x-x_{d}$ represents the control error, and λ is a strictly positive constant. Figure 2 presents the block diagram of the proposed intelligent control scheme.

Applying the control law (Equation 7) to the assumed plant dynamics (Equation 4), we obtain the following closed-loop system:(8)$$\dot{\tilde{x}}+\lambda \tilde{x}=d-\hat{d}$$

Equation (Equation 8) shows that the error dynamics is constantly driven by the approximation error $d-\hat{d}$. Nevertheless, as RBF networks are universal approximators, an arbitrary precision ε can be ensured for the uncertainty/disturbance estimation:(9)$$d={\hat{d}}^{*}+\varepsilon$$
with ${\hat{d}}^{*}$ being the optimal approximation related to the optimal weight vector $w^{*}$.

Let us evaluate the stability of the proposed control approach by defining positive definite Lyapunov candidate function *V*:(10)$$V=\frac{1}{2}{\tilde{x}}^{2}+\frac{1}{2\eta }\delta^{⊤}\delta$$
where η is a strictly positive constant and $\delta =w-w^{*}$.

Hence, since $\dot{\delta }=\dot{w}$, the time derivative of *V* is:$$\begin{array}{cc} \dot{V} & =\tilde{x}\;\dot{\tilde{x}}+\eta^{-1}\delta^{⊤}\dot{w}\\ \; & =\tilde{x}\left(-\lambda \;\tilde{x}+d-\hat{d}\;\right)+\eta^{-1}\delta^{⊤}\dot{w}\\ \; & =\tilde{x}\left(-\lambda \;\tilde{x}+\varepsilon +{\hat{d}}^{*}-\hat{d}\;\right)+\eta^{-1}\delta^{⊤}\dot{w}\\ \; & =\tilde{x}\left[-\lambda \;\tilde{x}+\varepsilon -{\left(w-w^{*}\right)}^{⊤}\varphi \left(\tilde{x}\right)\;\right]+\eta^{-1}\delta^{⊤}\dot{w}\\ \; & =\tilde{x}\left[-\lambda \;\tilde{x}+\varepsilon -\delta^{⊤}\varphi \left(\tilde{x}\right)\;\right]+\eta^{-1}\delta^{⊤}\dot{w}\\ \; & =-\tilde{x}\left(\lambda \;\tilde{x}-\varepsilon \;\right)+\eta^{-1}\delta^{⊤}\left[\dot{w}-\eta \tilde{x}\varphi \left(\tilde{x}\right)\;\right] \end{array}$$

If the weight vector is updated according to the following learning rule
(11)$$\dot{w}=\eta \;\tilde{x}\;\varphi \left(\tilde{x}\right)$$
then the time derivative of *V* becomes
(12)$$\dot{V}=-\tilde{x}\left(\lambda \;\tilde{x}-\varepsilon \;\right)$$

Equation (Equation 12) shows that $\dot{V}$ is negative semi-definite only for $|\tilde{x}|>\epsilon /\lambda$, with ϵ≥|ε| being the upper limit of the approximation error. In order to ensure an upper bound on w when $|\tilde{x}|\leq \epsilon /\lambda$, the projection algorithm [47] can be applied. Thus, the new learning rule becomes
(13)$$\dot{w}=\left\{\begin{array}{cc} \eta \;\tilde{x}\;\varphi \left(\tilde{x}\right) & {\mathrm{if}\;∥w∥}_{2}<\vartheta \;\mathrm{or}\\ \; & {\mathrm{if}\;∥w∥}_{2}=\vartheta \;\mathrm{and}\;\\ \; & \eta \;\tilde{x}\;w^{⊤}\varphi \left(\tilde{x}\right)<0\\ \left(I-\frac{ww^{⊤}}{w^{⊤}w}\right)\eta \;\tilde{x}\;\varphi \left(\tilde{x}\right) & \mathrm{otherwise} \end{array}\right.$$
with ϑ being the desired upper bound for ${∥w∥}_{2}$ (the Euclidian norm of w).

In fact, by adopting (Equation 13) we ensure that the weight vector always remains within the convex region $W=\{w\in R^{n}\;|\;w^{⊤}w\leq \vartheta^{2}\}$, as long as ${∥w\left(0\right)∥}_{2}\leq \vartheta$, and the control error converges to the closed region $X=\{\tilde{x}\in R\;|\;|\tilde{x}|\leq \epsilon /\lambda \}$ as t→∞.

## 4. Materials and Methods

The intelligent controller was evaluated through an in silico analysis using the Identifiable Virtual Patient model. The IVP model was implemented in C++ and numerically solved with the fourth-order Runge-Kutta method and a time step of $10^{-3}$ min. Three different scenarios were considered: one long- and two short-term simulation studies. In the short-term analyses, 20 virtual patients were simulated for a period of 7 days, while in the long-term simulation, 1 virtual patient was assessed over 63 days.

It is important to stress that the IVP model was considered for simulation purposes only and was not taken into account in the proposed control scheme design. As a matter of fact, the IVP model is represented by a four-dimensional nonlinear system (Equation 1) whereas the intelligent controller is designed for first-order systems like (Equation 4). It means that the controller only needs to measure one of the states of the IVP model, namely the blood glucose concentration G(t), so there is no need to estimate non-measurable states, such as the subcutaneous and plasma insulin concentrations $I_{\mathrm{SC}}\left(t\right)$ and I(t), respectively, nor the insulin effect on the blood glucose $I_{\mathrm{EFF}}\left(t\right)$. This allows the controller to rely solely on measurements from a blood glucose monitoring device, but requires it to be able to handle the unmodeled insulin dynamics and its effect on glucose metabolism.

The parameters for the IVP model were obtained from [58] and are summarized in Table 1. To take interpatient variability into account in the simulation, each patient had its parameters initialized from samples of normal distributions. The standard deviation for each distribution was chosen as 5% with respect to the corresponding mean value. Furthermore, to assess the controller’s efficacy against intrapatient variability, all model parameters were chosen to vary throughout the simulations. The parameters varied sinusoidally by ±10% over the average value with a period of one day. Both inter and intrapatient variability was considered to make the simulations more realistic and to assess the robustness of the proposed control scheme.

In order to appraise the capacity of the controller to cope with disturbances, food intake was considered three times a day: breakfast, lunch and dinner. Meal times and amounts of carbohydrate ingested ($C_{H}$) varied according to normal distributions, whose means and standard deviations are shown in Table 2.

Regarding the control scheme, it was considered that the blood glucose concentration could be measured once every 5 min, as usual in commercial CGM devices. Therefore, the control signal was updated using the same sampling period. The target glycemia was chosen as $x_{d}=110$ mg/dL, the control gain set as λ=1, and the upper bound for ${∥w∥}_{2}$ was defined as $\vartheta =3\times 10^{4}$. Moreover, as we propose a model-free approach, we assume that the controller designer has no prior knowledge about the plant dynamics and set $\hat{f}=0$ and $\hat{b}=1$. By choosing the identity elements of addition/subtraction and multiplication/division to define $\hat{f}$ and $\hat{b}$, respectively, we guarantee that none of them exerts any influence on the control signal, letting the plant dynamics be fully compensated by the neural network output $\hat{d}$.

For the RBF network, a single neuron was considered for the input layer, namely the control error $\tilde{x}$, and eleven neurons were chosen for the hidden layer. The activation functions were of the Gaussian type:(14)$$\varphi \left(\tilde{x}\right)=\exp\left[-\frac{1}{2}{\left(\frac{\tilde{x}-c}{l}\right)}^{2}\right]$$
with corresponding centers *c* and widths *l* being defined as in Table 3.

Two different protocols were adopted to adjust the neural network weights in the short-term simulation studies.

First, we let the neural network learn about the insulin/glucose interaction from scratch with the weight vector being initialized as $w_{0}=0$ and updated according to Equation (Equation 13). The learning rate of the neural network varied over time. We opted for a higher value for η in the first days, so that the neural network could recognize insulin action and glucose metabolism, as well as the dietary pattern of each individual. After the first week, a low learning rate was maintained in order to accommodate small metabolic fluctuations and food intake variations that might occur. The exact value of the learning rate adopted in the simulations is presented in Equation (Equation 15).
(15)$$\eta =\left\{\begin{array}{cc} 0.50, & \mathrm{on}\;\mathrm{day}\;1\\ 0.10, & \mathrm{on}\;\mathrm{days}\;2\;\mathrm{and}\;3\\ 0.05, & \mathrm{from}\;\mathrm{day}\;4\;\mathrm{to}\;7\\ 0.01, & \mathrm{from}\;\mathrm{day}\;8\;\mathrm{onwards} \end{array}\right.$$

Then, in order to assess how the proposed algorithm would behave in a transition from a different therapy protocol, we adopted a hybrid approach. We assumed that patients were being treated with continuous basal insulin therapy using their automated delivery systems. During the last three days of basal therapy, we collected the patients’ blood glucose levels along with the corresponding infusion rates, and used this data to train the neural network offline using the standard pseudoinverse matrix. Thus, when switching to the therapy based exclusively on the intelligent controller on the fourth day, the weight vector w was initialized with the values calculated from the pseudoinverse matrix and thereafter updated according to Equation (Equation 13). The learning rates adopted in this case are presented in Equation (Equation 16).
(16)$$\eta =\left\{\begin{array}{cc} 0.50, & \mathrm{on}\;\mathrm{day}\;4\\ 0.10, & \mathrm{on}\;\mathrm{days}\;5\;\mathrm{and}\;6\\ 0.05, & \mathrm{on}\;\mathrm{day}\;7\\ 0.01, & \mathrm{from}\;\mathrm{day}\;8\;\mathrm{onwards} \end{array}\right.$$

It is also important to highlight, for the evaluation of glycemic control, standardized CGM metrics for clinical care [59,60] were used. In this way, the values of mean BGC, standard deviation (SD), coefficient of variation (CV) and time in range are calculated for the simulations. For the coefficient of variation, the target range is below 36%, since values above this limit are commonly associated with higher risks of hyperglycemia and higher glycemic variability [61].

## 5. Results and Discussion

First, the results obtained in the short-term simulations (20 patients for 7 days with 3 meals a day) and with the weight vector initialized as $w_{0}=0$ are presented in Figure 3 and Figure 4 as well as in Table 4.

Figure 3 shows the mean glycemia and the relative standard deviation obtained from the 20 patients over 7 days. The results demonstrate that, after a brief learning phase that lasted less than a day, the proposed intelligent controller was able to maintain the patients’ normoglycemia for the remaining six days. During the first day, the learning rate of the neural network was set to its highest value η=0.5, so that the metabolism of the glucose-insulin interaction could be properly estimated. In fact, this is what caused the brief hyperglycemic peak at the very beginning of the controller’s actuation. Although none of the simulated patients presented severe hypoglycemia (below 54 mg/dL) or acute hyperglycemia (above 400 mg/dL), based on the results obtained, it would be recommended that real patients be followed up by a healthcare professional during the neural network’s learning phase.

As a matter of fact, if the first three days (when the learning rates were higher, i.e., η≥0.10) are not considered in the analysis, then we can assert that none of the patients had hypoglycemia (whether severe or moderate) or severe hyperglycemia. On average, patients had a time in range (TIR) of 96.76% (normoglycemia) and a time above range (TAR) of 3.24% (moderate hyperglycemia). This confirms the great performance of the proposed intelligent controller, since the targets for assessment of glycemic control in patients with diabetes are, respectively, TIR>70% and TAR<25% for moderate hyperglycemia [59,60].

In addition, in order to have a glimpse of the extreme blood glucose concentrations in the in silico analysis, the maximum and minimum glycemic values (peak, mean, standard deviation, and coefficient of variation) obtained in the simulations are presented in Table 4. These results are from the period after the initial learning phase (three days) and the corresponding patients are shown in parentheses. It is noteworthy that the coefficient of variation (CV) remained below the target of 36% for all patients. The highest coefficient of variation (obtained for patient 1) was in fact 30.62%.

Regarding the insulin infusion, Figure 4 presents the mean values and their relative standard deviation obtained from the 20 patients. The main fluctuations in the insulin infusion that typically occur three times a day, as can be clearly seen in the figure, are caused by food intake. It is important to emphasize that the control system was not previously informed about the meals or the amount of carbohydrate, which attests to its efficacy in recognizing the food intake and automatically compensating for it.

Now consider the case of patients with ongoing basal therapies who have switched to a protocol based on the intelligent control scheme. Figure 5 and Figure 6 , as well as Table 5, present the results obtained for a protocol of 50 U of insulin per day during the three days prior to the outset of the intelligent control approach.

As observed in Figure 5, the adoption of a continuous basal therapy before starting the intelligent control scheme can be very useful to prevent severe hyperglycemic events. Continuous basal therapy is a commonly used approach at the beginning of insulin-based treatment [62], but due to its own limitations [63,64], it is generally not able to maintain the desired level of glycemic control over the long term. However, if carried out with adequate medical supervision, it can help in the prior training of the neural network. In fact, thanks to the offline adjustment of the weights performed in the transition from continuous basal to intelligently regulated therapy (from the third to the fourth day), it was possible to maintain the blood glucose level within a much safer glycemic range (Figure 5), when compared to the approach in which ANN had to learn from scratch (Figure 3).

Figure 6 shows that the intelligent control scheme is able to adjust itself more quickly at the beginning of the treatment, when compared to the case without previous training Figure 4. Moreover, by regulating insulin infusion according to demand throughout the day, the intelligent controller is able to automatically handle both inter and intrapatient variability, which is not the case with basal therapy employed in the training phase.

After three days of basal therapy, patients had an average time in the range (TIR) of 96.73% (normoglycemia), a time above range (TAR) of 2.77% (moderate hyperglycemia), and a time below range (TBR) of 0.50% (moderate hypoglycemia), which represent very adequate metrics when compared to their targets, respectively, TIR>70%, TAR<25% for moderate hyperglycemia, and TBR<4% for moderate hypoglycemia [59,60]. In addition, Table 5 provides a summary of extreme blood glucose concentrations after basal therapy. It is worth mentioning that also in this case all the coefficients of variation (CV) were below the target of 36%, with the highest value associated with patient 1 (32.16%).

Regarding the long-term analysis, the results obtained for a virtual patient considering 63 simulated days and 3 meals a day are shown in Figure 7 and Figure 8. These results are presented in order to emulate an ambulatory glucose profile (AGP) and have been depicted on a 24-h basis. The AGP represents a standardized report in which daily glucose and insulin profiles are qualitatively and quantitatively summarized. It is an internationally recognized standard for reporting CGM data [59] and allows for easy identification of glycemic median, variability (using interpercentile ranges), mean, standard deviation, coefficient of variation, and time in ranges. As the objective now is to identify the effectiveness of the proposed control scheme in the long term, data from the initial learning phase (first three days) of the neural network were not included in this analysis.

As can be seen in Figure 7, the proposed intelligent controller was able to maintain normoglycemia with an associated time in range of 95.19%. Furthermore, there were no episodes of hypoglycemia, whether severe or moderate, nor severe hyperglycemia. Moderate hyperglycemia was observed with a corresponding time above range of 4.81%, which is well below the acceptable limit of TAR<25% [59,60]. Mean glycemia was 119.59 mg/dL with a standard deviation (SD) of 32.02 mg/dL and a coefficient of variation (CV) of 26.78%, both below the reference values [61] of SD < 50 mg/dL and CV < 36%, respectively.

In addition, fluctuations in both glycemia (see Figure 7) and insulin infusion (see Figure 8) after meals can be easily verified. These long-term results confirm that the increase in blood glucose concentration due to food intake is readily associated with an effective controller response, which delivers higher doses of insulin to counteract the effects of the ingested carbohydrate.

## 6. Concluding Remarks

This work introduces a novel intelligent control scheme for automated insulin delivery systems (also known as an artificial pancreas). By combining an adaptive neural network with a Lyapunov-based method, the proposed model-free controller is able to identify the physiological characteristics of the glucose/insulin interaction, as well as the dietary pattern of each patient, and automatically regulate the insulin infusion. Moreover, the control scheme only needs to know the glycemia every five minutes, not requiring non-measurable variables related to insulin dynamics and insulin-glucose metabolism. In order to ensure patient safety, the stability of the control law is rigorously addressed through a Lyapunov-like analysis. The ability of the intelligent controller to maintain normoglycemia is confirmed through in silico analysis using virtual patients. The obtained results demonstrate that the proposed scheme is able to deal with inter- and intrapatient variability and does not require any information about food intake or amount of carbohydrate ingested. Finally, it is also important to note that by combining the artificial neural networks with a nonlinear control method, the computational demand on the ANN is drastically reduced, which allows its implementation in the embedded hardware of AID systems.

## Figures and Tables

**Figure 1 bioengineering-09-00664-f001:**
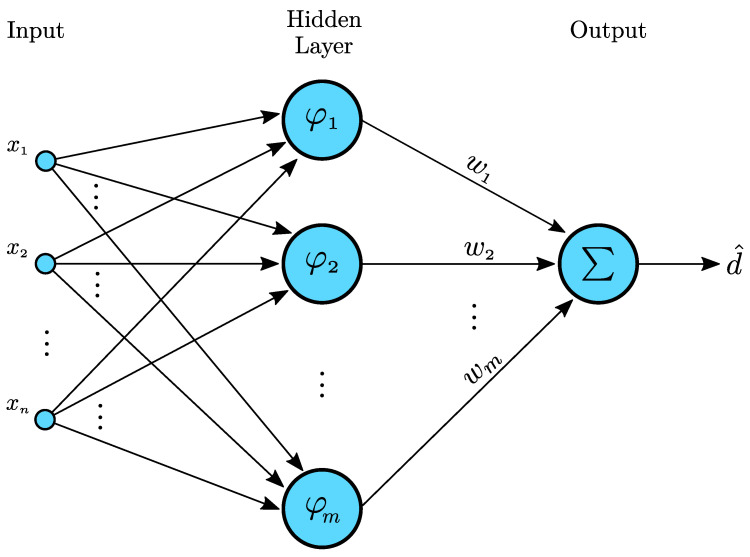
Architecture of a RBF neural network.

**Figure 2 bioengineering-09-00664-f002:**
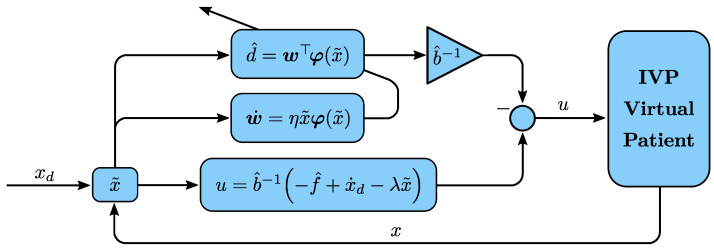
Block diagram of the proposed intelligent controller.

**Figure 3 bioengineering-09-00664-f003:**
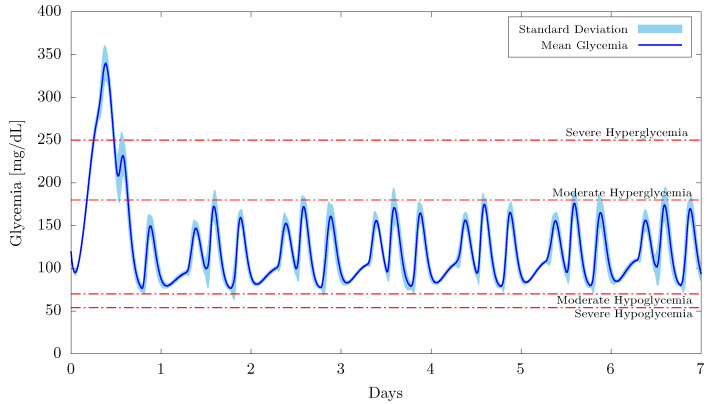
Mean glycemia and standard deviation for 20 patients and considering $w_{0}=0$.

**Figure 4 bioengineering-09-00664-f004:**
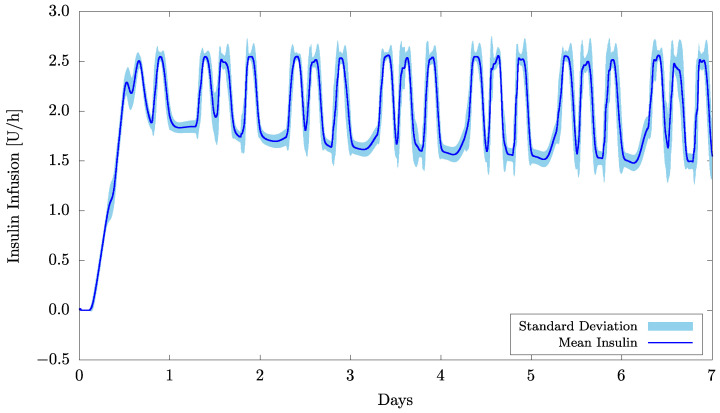
Mean insulin infusion and standard deviation for 20 patients and considering $w_{0}=0$.

**Figure 5 bioengineering-09-00664-f005:**
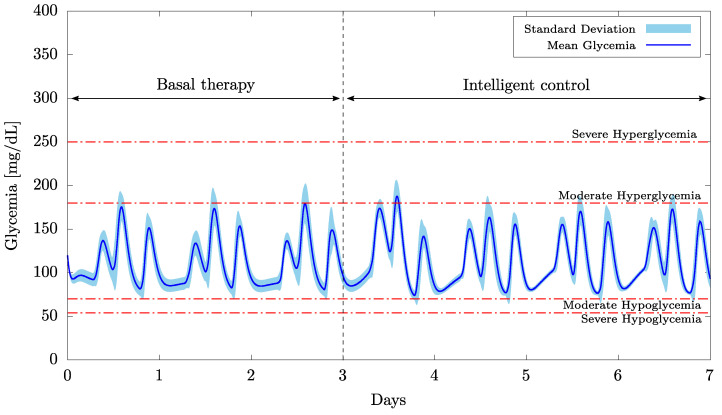
Mean glycemia and standard deviation for 20 patients with prior basal therapy.

**Figure 6 bioengineering-09-00664-f006:**
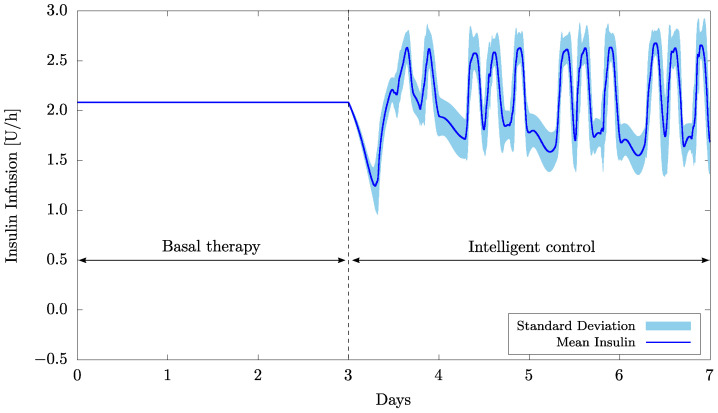
Mean insulin infusion and standard deviation for 20 patients with prior basal therapy.

**Figure 7 bioengineering-09-00664-f007:**
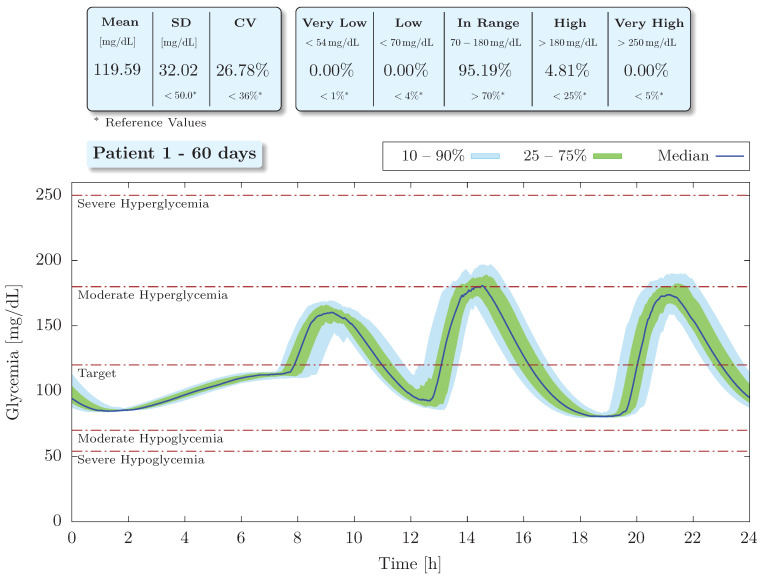
Glycemic profile of patient 1 with median and interpercentile ranges.

**Figure 8 bioengineering-09-00664-f008:**
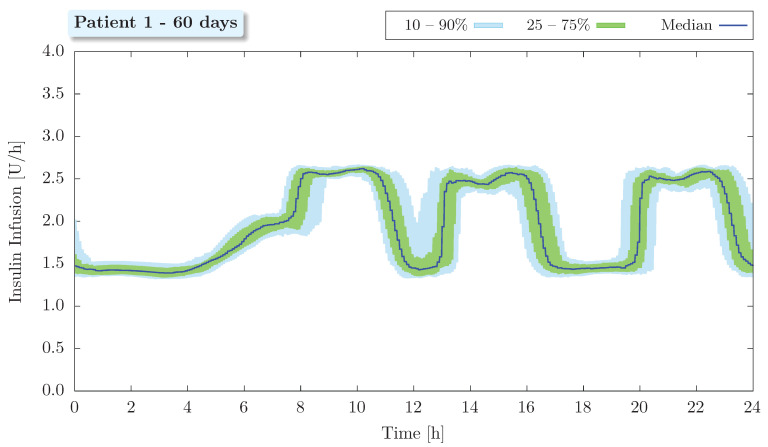
Insulin profile of patient 1 with median and interpercentile ranges.

**Table 1 bioengineering-09-00664-t001:** Mean values adopted to generate the parameters for virtual patients with the IVP model.

	$\tau_{1}$	$\tau_{2}$	$C_{I}$	$p_{2}$	$S_{I}$	GEZI	EGP	$\tau_{m}$	$V_{G}$
Mean	49.00	47.00	2010.00	1.06 $\times \;10^{-2}$	$8.11\times 10^{-4}$	$2.20\times 10^{-3}$	1.33	40.50	253.00

**Table 2 bioengineering-09-00664-t002:** Time and amount of carbohydrate of each meal.

	Breakfast	Lunch	Dinner
Time (Mean ± SD) [hh:mm]	07:30 ± 00:30	12:30 ± 00:30	19:30 ± 00:30
Carbohydrate (Mean ± SD) [g]	42.00 ± 3.50	66.00 ± 5.50	51.00 ± 4.25

**Table 3 bioengineering-09-00664-t003:** Parameters of the Gaussian-type activation functions.

	$\varphi_{1}$	$\varphi_{2}$	$\varphi_{3}$	$\varphi_{4}$	$\varphi_{5}$	$\varphi_{6}$	$\varphi_{7}$	$\varphi_{8}$	$\varphi_{9}$	$\varphi_{10}$	$\varphi_{11}$
Center (c)	−25.0	−10.0	−8.0	−5.0	−2.0	0.0	5.0	10.0	20.0	40.0	100.0
Width (l)	20.0	15.0	10.0	8.0	5.0	5.0	5.0	20.0	40.0	100.0	200.0

**Table 4 bioengineering-09-00664-t004:** Glycemic metrics considering 20 patients and without prior basal therapy.

	Peak [mg/dL]	Mean [mg/dL]	SD [mg/dL]	CV [%]
Maximum	213.04 (Patient 1)	119.33 (Patient 7)	35.89 (Patient 1)	30.62 (Patient 1)
Minimum	74.77 (Patient 16)	113.45 (Patient 20)	26.47 (Patient 15)	23.21 (Patient 15)

**Table 5 bioengineering-09-00664-t005:** Glycemic metrics considering 20 patients and with prior basal therapy.

	Peak [mg/dL]	Mean [mg/dL]	SD [mg/dL]	CV [%]
Maximum	221.51 (Patient 1)	117.33 (Patient 19)	36.27 (Patient 1)	32.16 (Patient 1)
Minimum	59.91 (Patient 7)	107.74 (Patient 11)	28.56 (Patient 12)	25.06 (Patient 12)

## Data Availability

Not applicable.

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
