# Peer review of "Intelligent Control with Artificial Neural Networks for Automated Insulin Delivery Systems"

_bioengineering, 2022, doi:10.3390/bioengineering9110664_

Round 1

Reviewer 1 Report

This manuscript presents a model-free adaptive control approach to insulin dosing. It addresses a significant health problem. The material is well presented and easy to follow. 

The proposed controller is a state feedback controller designed using Lyapunov stability theory. In general, this approach is not necessarily novel in the context of automated insulin dosing: (Greenwood & Gunton, JDST 2014), (Sepasi et al. Biomedical Sign Proc. and Control, 2021), (Toffanin et al, IFAC Proceedings, 2014).   RBF network is used as a function approximator for estimating the disturbance due to meal intake.  During adaptation phase the basis functions remain fixed and the weights are updated based on the proposed Lyapunov stability function.   

Question to the authors: "Moreover, as we assume that the plant dynamics is totally unknown, we chose ˆ f = 0 and ˆb = 1."  Since this approach is model-free, why do we even have to make this assumption ?

The results show the initial feasibility of the proposed method. It would be really interesting to see what happens if in the initial (adaptation / training) phase a different, already established, insulin dosing algorithm is used while the RBF weights are adjusted. This may prevent the hyperglycemic episode observed. One could imagine that most patients would be transitioning from another insulin dosing algorithm. An interesting simulation idea would be to show that transition process and the resulting change in CGM metrics.

Reviewer 2 Report

The paper is well written. The conclusion is clear. I have the following comments.

1.      Abstract Section needs to be rewritten to highlight the contribution of the study and to report the numerical results of the study.

2.      A comparison with the literature studies is required.

3.      A figure for the whole framework is needed to show the components and how they are related.

4.     Results should be organized in tables and figures, and the proposed model should be compared with other machine learning models.
